# Mechanistic insights into the protective roles of polyphosphate against amyloid cytotoxicity

Justine Lempart[1,2], Eric Tse[3], James A Lauer[2], Magdalena I Ivanova[4,5] , Alexandra Sutter[4] , Nicholas Yoo[2], Philipp Huettemann[2], Daniel Southworth[3], Ursula Jakob[2,6]

The universally abundant polyphosphate (polyP) accelerates fibril formation of disease-related amyloids and protects against amyloid cytotoxicity. To gain insights into the mechanism(s) by which polyP exerts these effects, we focused on α-synuclein, a well-studied amyloid protein, which constitutes the major component of Lewy bodies found in Parkinson's disease. Here, we demonstrate that polyP is unable to accelerate the rate-limiting step of α-synuclein fibril formation but effectively nucleates fibril assembly once α-synuclein oligomers are formed. Binding of polyP to α-synuclein either during fibril formation or upon fibril maturation substantially alters fibril morphology and effectively reduces the ability of α-synuclein fibrils to interact with cell membranes. The effect of polyP appears to be α-synuclein fibril specific and successfully prevents the uptake of fibrils into neuronal cells. These results suggest that altering the polyP levels in the extracellular space might be a potential therapeutic strategy to prevent the spreading of the disease.

## Introduction

Parkinson's disease (PD), the second most common neurodegenerative disorder known (de Lau et al, 2004), is characterized by a loss of dopaminergic neurons in the substantia nigra (Rinne et al, 1989; Damier et al, 1999). A hallmark of the disease is the appearance of intracellular protein inclusions (i.e., Lewy bodies), which consist primarily of insoluble fibrils of α-synuclein, a 140–amino acid protein involved in presynaptic vesicle formation (Goedert, 2001; Shults, 2006). Although it is now well established that deposition of α-synuclein fibrils associates with the disease and that cell death can be elicited simply by incubating neuronal cells with α-synuclein fibrils (Winner et al, 2011), many open questions remain concerning the mechanism of toxicity, the

structural features of the toxic α-synuclein species, and the way(s) by which α-synuclein toxicity propagates in the brain.

In solution, α-synuclein is a soluble monomer with extensive regions of intrinsic disorder (Fakhree et al, 2018). In vitro studies demonstrated that upon prolonged incubation, α-synuclein monomers undergo conformational rearrangements, which lead to the formation of aggregation-sensitive oligomers (Conway et al, 2000b). These nuclei are capable of sequestering other α-synuclein monomers and will grow into protofibrils and eventually into insoluble, protease-resistant fibrils (Wood et al, 1999; Conway et al, 2000a). In vitro, the rate-limiting step in fibril formation appears to be the formation of the initial nuclei, and fibril formation has been shown to be accelerated by the addition of negatively charged polymers, including glucosaminoglycans (i.e., heparin) (Cohlberg et al, 2002), RNA (Munishkina et al, 2009), or phospholipids (Zhu et al, 2003). The precise roles that these additives play in in vivo fibril formation remain to be determined.

Recent studies provided supporting evidence that amyloid toxicity is not caused by the fibrils per se but by oligomeric species that transiently accumulate on the pathway to fibril formation (Winner et al, 2011; Chen et al, 2015). These oligomers, which have been shown to affect mitochondrial function (Luth et al, 2014), membrane permeability (Lashuel et al, 2002; Tsigelny et al, 2012), and/or the cytoskeleton (Roberts & Brown, 2015), are thought to be responsible for the observed neuroinflammation (Lee et al, 2010) and cell death (Winner et al, 2011). Moreover, amyloid oligomers seem to be the primary species that spread among cells (Danzer et al, 2009, 2012) and to be responsible for the prion-like propagation of PD pathology (Li et al, 2008; Desplats et al, 2009). Cell-to-cell transmission appears to involve the active secretion of α-synuclein oligomers into the extracellular space followed by the uptake of the amyloids into neighboring recipient cells via micropinocytosis and glycosaminoglycan receptors (Holmes et al, 2013; Reyes et al, 2015; Gustafsson et al, 2018). Experiments conducted in cell culture confirmed that α-synuclein oligomers can readily spread between neurons and glial cells (Hansen et al, 2011; Domert et al, 2016), and, once taken up by recipient cells, sequester monomeric α-synuclein into insoluble foci (Reyes et al, 2015; Rostami et al, 2017).

[1]Graduate Program in Biochemistry, Department of Chemistry, Technische Universität München, München, Germany   [2]Department of Molecular, Cellular and Developmental Biology University of Michigan, Ann Arbor, MI, USA   [3]Institute for Neurodegenerative Diseases, Department of Biochemistry and Biophysics, University of California, San Francisco, CA, USA   [4]Biophysics Program, University of Michigan, Ann Arbor, MI, USA   [5]Department of Neurology, University of Michigan, Ann Arbor, MI, USA   [6]Department of Biological Chemistry, University of Michigan, Ann Arbor, MI, USA

Correspondence: ujakob@umich.edu

Recent work from our laboratory demonstrated that polyphosphate (polyP), a highly conserved and universally present polyanion, significantly decreases the cytotoxicity of amyloidogenic proteins (Cremers et al, 2016). These results were corroborated in studies with amyloid $\beta_{25–35}$, which showed that preincubation of PC12 cells or primary cortical neurons with polyP protects against the neurotoxic effects of the peptide (Muller et al, 2017). In vitro studies revealed that polyP substantially accelerates $\alpha$-synuclein fibril formation in a chain length–dependent manner, causing the formation of both shedding-resistant and seeding-deficient polyP-associated fibrils (Cremers et al, 2016). Localization studies revealed that polyP, such as $\alpha$-synuclein, is both secreted and taken up by neuronal cells and, hence, localizes both inside and outside of cells (Angelova et al, 2018). These results raised intriguing questions as to what $\alpha$-synuclein species interact with polyP, how premature fibril formation might be avoided, and, most importantly, by what mechanism polyP is able to protect neuronal cells against $\alpha$-synuclein toxicity.

Here, we show that polyP does not interact with monomeric $\alpha$-synuclein but effectively nucleates $\alpha$-synuclein fibril formation once prefibrillar species are present. PolyP causes pronounced morphological changes in both de novo forming fibrils as well as upon its addition to mature $\alpha$-synuclein fibrils, demonstrating that $\alpha$-synuclein fibrils are inherently dynamic and amendable to polyP-mediated structural changes. Importantly, presence of polyP strongly interferes with the interaction of $\alpha$-synuclein fibrils with cell membranes and prevents the uptake of $\alpha$-synuclein fibrils into differentiated neuroblastoma cells. These results explain the cytoprotective effect of polyP and suggest that extracellular polyP might be able to influence the spreading of this disease.

# Results

## PolyP accelerates fibril formation by nucleating $\alpha$-synuclein oligomers

Amyloid fibril formation is most commonly monitored by measuring the fluorescence of thioflavin T (ThT), a small molecular dye that becomes highly emissive when intercalated into the $\beta$-sheets of amyloidogenic oligomers and fibrils (LeVine, 1999). ThT kinetics of amyloid fibril formation can be divided into three distinct phases (Fig 1A): the nucleation (i.e., lag) phase, in which soluble monomers undergo structural changes and nucleate; the elongation (i.e., growth) phase, during which ThT-positive oligomers and protofibrils form; and the equilibration (i.e., plateau) phase, in which mature fibrils undergo cycles of shedding and seeding (Shoffner & Schnell, 2016).

Consistent with previous polyP titrations (Cremers et al, 2016), we found that the presence of 500 $\mu$M polyP substantially accelerates

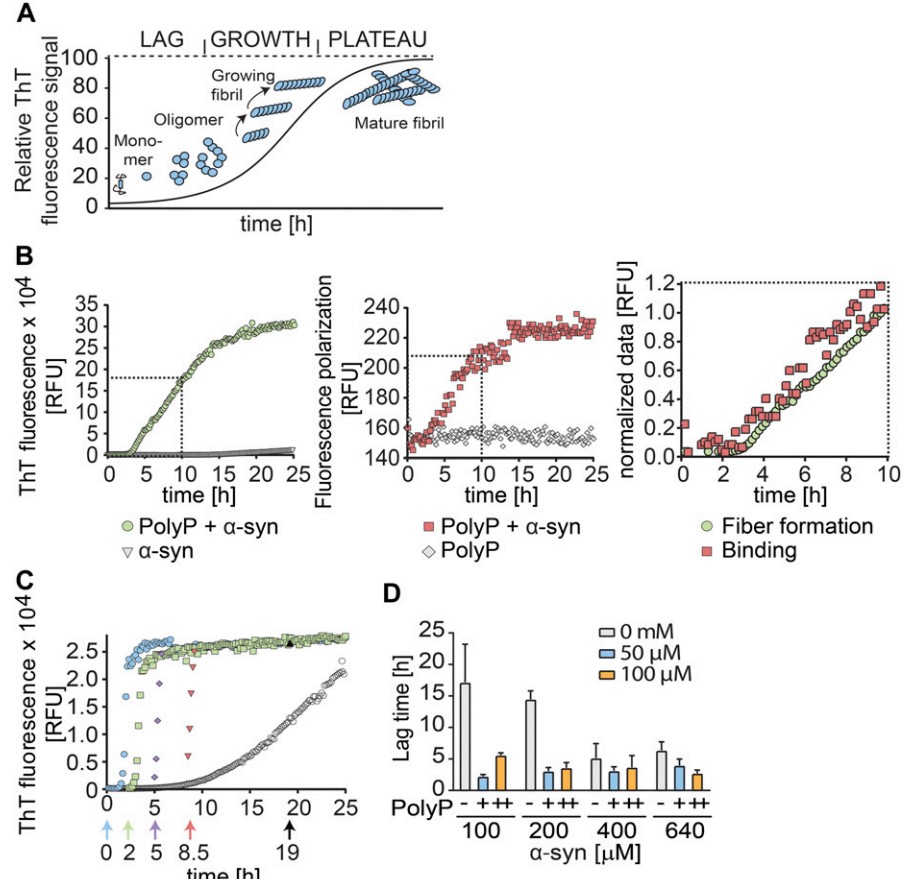

**Figure 1. Influence of polyP on $\alpha$-synuclein fibril formation in vitro.**
**(A)** Model of the amyloid-fibril forming process using ThT-fluorescence. **(B)** 100 $\mu$M freshly prepared $\alpha$-synuclein was incubated in the absence or presence of 500 $\mu$M fluorescently labeled polyP$_{300\text{-AF647}}$ (in P$_i$ units) at 37°C under constant stirring. ThT fluorescence was used to monitor fibril formation (left panel), and FP experiments were conducted to measure binding of polyP$_{300\text{-AF647}}$ to $\alpha$-synuclein (middle panel). Overlay of normalized ThT and FP-curves (right panel). Data are the mean of three independent experiments ±SD. **(C)** Addition of 500 $\mu$M polyP (in P$_i$ units) before (cyan circles) or at defined time points during the fibril-forming process of 300 $\mu$M $\alpha$-synuclein. ThT fluorescence was monitored. All experiments were conducted at least three times. Representative kinetic traces are shown. **(D)** Influence of different polyP$_{300}$ and $\alpha$-synuclein concentrations on the lag phase of fibril formation. ThT fluorescence was monitored and the lag phase was determined. The mean of four experiments ±SD is shown.
Source data are available for this figure.

the fibril-forming process of $\alpha$-synuclein, both by shortening the lag phase and by increasing the rate of fibril growth (Fig 1B and C). To determine when during the polymerization process polyP acts on amyloidogenic proteins, we conducted fluorescence polarization (FP) measurements, which record the tumbling rate of fluorescent molecules as readout for real-time binding events (Rossi & Taylor, 2011). We labeled polyP$_{300}$-chains (molecular weight [MW]: ~30 kD) with Alexa Fluor 647 (polyP$_{300\text{-}AF647}$) and conducted FP measurements in the presence of freshly prepared $\alpha$-synuclein (MW: 14 kD) for 40 h (Fig 1B, middle panel). Unexpectedly, we did not observe any significant increase in the FP signal over the first ~2.5 h of incubation, suggesting that polyP does not interact with monomeric $\alpha$-synuclein. After this lag-phase, however, the FP signal rapidly increased and reached an apparent plateau after about 12 h of incubation (Fig 1B, middle panel). Because we could not exclude that the plateau was not due to the size of the polyP–fibril complexes reaching the upper limit of our polarization measurements, we only directly compared the first 10 h of ThT binding and anisotropy measurements. Overlay of the normalized data revealed that although the lag phase in the FP measurement was slightly shorter than the lag phase in the ThT measurements, the rate of signal increase was very similar (Fig 1B, right panel). These results suggested that polyP does not interact with $\alpha$-synuclein species that occur early in the fibril-forming process (i.e., monomers) but instead binds $\alpha$-synuclein species shortly before or concomitant with their ability to intercalate ThT.

Time-delayed polyP addition experiments confirmed these results and demonstrated that polyP acts on nucleation-competent oligomers and/or protofibrils. For these studies, we used experimental conditions under which $\alpha$-synuclein fibril formation proceeds with a lag phase of ~6 h in the absence and ~1.5 h in the presence of polyP (Fig 1C, compare open and cyan circles). When we added polyP 2 h after the start of the incubation, the lag phase was reduced from the remaining 4 h in the absence of polyP to less than 30 min (Fig 1C, green squares). Addition of polyP after 5 h caused an immediate increase in ThT signal (Fig 1C, blue diamonds), whereas addition of polyP mid-way through the elongation phase of $\alpha$-synuclein fibril formation triggered maximal ThT binding within less than 10 min (Fig 1C, red triangles). These results strongly suggested that polyP binds to a range of presumably non-monomeric $\alpha$-synuclein species and supports their association into insoluble fibrils.

## PolyP does not affect rate-limiting step of fibril formation

Our finding that polyP does not detectably interact with $\alpha$-synuclein monomers but readily stimulates fibril formation once ThT-positive oligomers have formed, suggested that polyP does not affect the rate-limiting step of $\alpha$-synuclein fibril formation. To test this idea, we combined increasing $\alpha$-synuclein concentrations with increasing polyP concentrations and measured the respective lag phase of fibril formation using ThT fluorescence (Fig 1D). As expected, increasing the $\alpha$-synuclein concentration from 100 to 400 $\mu$M in the absence of polyP reduced the lag phase from about 16 h to less than 7 h. Higher concentrations of $\alpha$-synuclein (i.e., 640 $\mu$M) did not significantly shorten the lag phase any further. The presence of physiological relevant concentrations of polyP$_{300}$ (50 $\mu$M in P$_i$-units) (Kumble & Kornberg, 1995; Holmstrom et al, 2013) reduced this lag time to 2–3 h. Noteworthy, this reduction in lag time

appeared to be independent of the $\alpha$-synuclein concentration used (Fig 1D). Moreover, doubling the polyP concentration also failed to further reduce the lag phase. These results agreed with previous results showing that $\alpha$-synuclein undergoes conformational changes and/or oligomerization processes that are rate limiting (Wood et al, 1999; Krishnan et al, 2003) and suggested that this step cannot be accelerated by the presence of polyP. We concluded from these results that simple co-existence of polyP and $\alpha$-synuclein in the same (extra)cellular compartment will unlikely be sufficient to trigger de novo fibril formation.

## PolyP alters morphology of preformed $\alpha$-synuclein fibrils

FP-binding studies using preformed $\alpha$-synuclein fibrils revealed that polyP not only interacts with ThT-positive oligomers during de novo fibril formation but also binds to mature fibrils (Fig 2A). Because $\alpha$-synuclein fibrils that are formed in the presence of polyP (i.e., $\alpha$-syn$^{polyP}$) have significantly altered morphology compared with fibrils formed in the absence of nucleators (i.e., $\alpha$-syn$^{alone}$) (Cremers et al, 2016), we wondered whether polyP binding would also affect the morphology of mature fibrils. This would possibly explain why the addition of polyP to preformed fibrils was as cytoprotective as its addition during fibril formation (Cremers et al, 2016). We, therefore, generated $\alpha$-synuclein fibrils, washed and purified them to remove any small oligomers and protofibrils, and either left them untreated ($\alpha$-syn$^{alone}$) or incubated them with polyP$_{300}$ ($\alpha$-syn$^{alone \rightarrow polyP}$). Immediately before as well as 20 min after the addition of polyP to $\alpha$-syn$^{alone}$ fibrils, we fixed aliquots of the samples on grids and prepared them for transmission electron microscopy (TEM). As a control, we also tested $\alpha$-synuclein fibrils formed in the presence of polyP ($\alpha$-syn$^{polyP}$). As shown in Fig 2B (Fig S1), the morphology of $\alpha$-syn$^{alone \rightarrow polyP}$ fibrils was nearly indistinguishable from the morphology of $\alpha$-syn$^{polyP}$ fibrils. Instead of two protofilaments, which typically form a twisted structure, $\alpha$-syn$^{alone \rightarrow polyP}$ and $\alpha$-syn$^{polyP}$ fibrils were significantly thinner, suggesting that polyP caused their dissociation into single protofilaments. X-ray fibril diffraction measurements agreed with the finding that incubation of preformed fibrils with polyP alters their conformations and showed particularly striking differences in the equatorial plots of the radial intensities (i.e., X-axis), which arise from the packing of adjacent $\beta$-sheets in the amyloid fibril. In contrast, no differences were observed on the meridian (Y-axis), which reflects the strand-to-strand packing, and produced a sharp reflection at 4.7 Å spacing for both fibril species (Fig 2C and D). These results suggested a pronounced effect of polyP on the packing of the $\beta$-sheets within the protofilament (Fig 2E).

To further investigate the dynamics of polyP–fibril interactions, we conducted FP competition experiments with preformed $\alpha$-syn-polyP$_{300\text{-}AF647}$ fibrils (Fig 2F). As expected, we observed a high initial FP signal, consistent with the slow tumbling rate of polyP–fibril complexes. Upon addition of unlabeled polyP$_{300}$, however, the FP signal rapidly decreased, indicating that the unlabeled polyP chains replaced the labeled polyP in the fibrils. Addition of the much shorter polyP$_{14}$ chain also reduced the FP signal but to a lesser extent, suggesting that shorter chains have lower binding affinities than longer chains (Fig 2F). These results indicated that the polyP–fibril interactions are highly dynamic in nature and implied that fibrils,

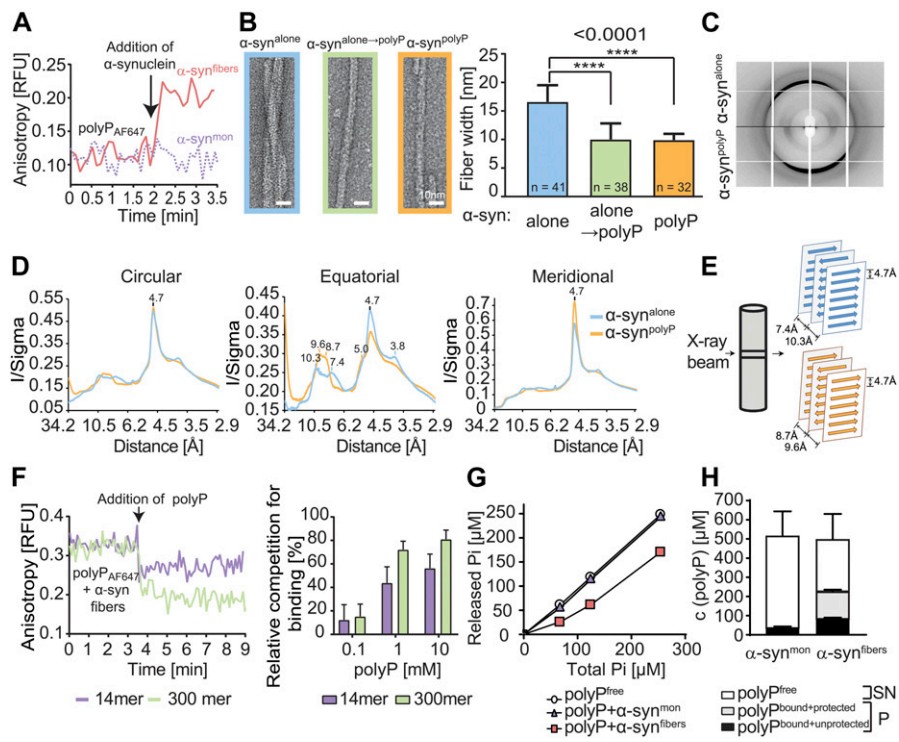

**Figure 2. Effects of polyP on α-synuclein fibril morphology.**
**(A)** FP of 50 μM polyP$_{300-AF647}$ upon addition of 30 μM α-syn$^{mon}$ or α-syn$^{fibrils}$. The arrow indicates the time point of protein addition. **(B)** TEM of α-synuclein fibrils (300 μM) formed in the absence of polyP and left untreated (α-syn$^{alone}$) or incubated with 7.5 mM polyP$_{300}$ for 20 min at RT (α-syn$^{alone \rightarrow polyP}$). α-synuclein fibrils formed in the presence of 7.5 mM polyP$_{300}$ were used as control (α-syn$^{polyP}$). Quantitative analysis of fibril width was based on 10 individual micrographs and 32–41 α-synuclein filaments per condition (additional images of representative fibers can be found in Fig S1). Statistical analysis was prepared with One-way ANOVA (****P-value < 0.0001). **(C, D)** X-ray fiber diffraction of α-synuclein formed in the absence or presence of polyP$_{300}$. The oriented samples produced cross-β diffraction patterns that contained a sharp reflection at 4.7 Å spacing at the meridian (Y-axis) and a broad reflection at ~9 Å spacing at the equator (X-axis) (C) The intensities were radially averaged over a full circle (360°, left panel), an equatorial arc (±30° around X-axis, middle panel), and meridional arc (±30° around Y-axis, right panel) (D). **(E)** Cartoon representation of the possible ways β-sheets and strands assemble in the fibril. **(F)** Left panel: FP of 30 μM of preformed α-synuclein–polyP$_{300-AF647}$ fibrils before and after the addition of 1 mM unlabeled polyP$_{300}$ or polyP$_{14}$. The arrow indicated the time point of polyP addition. Right panel: varying concentrations of polyP$_{14}$ or polyP$_{300}$ in the competition experiment. The percent competition was calculated from the relative signal change upon polyP addition, setting the polyP$_{300-AF647}$ fibril signal to 0% competition and the polyP$_{300-AF647}$ alone signal as 100% competition. The mean of three experiments ±SD is shown. **(G)** 40 μM α-synuclein monomers (triangles) or preformed fibrils (squares) were incubated with increasing concentrations of polyP$_{300}$ for 10 min. The samples were treated with ScPPX and assayed for released P$_i$. A standard curve of polyP$_{300}$ in the absence of α-synuclein (circles) was used as control. **(H)** 40 μM α-synuclein monomers or preformed fibrils were incubated with 500 μM polyP$_{300}$ for 10 min. The samples were separated into supernatant (SN) and pellet (P), treated with ScPPX, and subsequently assayed for P$_i$. PolyP$_{bound-unprotected}$ represents the amount of P$_i$ that was released upon ScPPX treatment in the pellet fraction. The amount of polyP not released by ScPPX was considered to be protected by the fibrils against hydrolysis (polyP$_{bound + protected}$).
Source data are available for this figure.

even when formed in the absence of polyP, can rapidly adopt a novel conformation when exposed to polyP.

## PolyP–fibril complexes are polyphosphatase resistant

Unbound polyP is very rapidly degraded by exopolyphosphatases, such as yeast polyphosphatase (PPX), which hydrolyzes the phosphoanhydride bonds with a turnover rate of 500 μmol/mg/min at 37°C (Wurst & Kornberg, 1994). To test whether degradation of polyP reverses the morphological changes that we observed in fibrils bound to polyP, we incubated α-syn$^{polyP}$ fibrils with yeast PPX. Surprisingly, however, we did not observe any morphological changes in the α-synuclein fibrils by TEM (data not shown). These results suggested either that the fibrils maintain their altered conformation even upon hydrolysis of polyP or that polyP, once in complex with fibrils, resists PPX-mediated hydrolysis. To investigate whether PPX is able to degrade fibril-associated polyP, we incubated 40 μM α-synuclein monomers or preformed α-synuclein fibrils with increasing concentrations of polyP, added PPX, and measured PPX-mediated release of P$_i$ using a modified molybdate assay (Christ & Blank, 2018). Whereas polyP that was incubated with α-synuclein monomers was rapidly hydrolyzed and yielded in the expected amount of P$_i$ (Fig 2G, triangles), presence of 40 μM α-synuclein fibrils protected about 130 μM of P$_i$ units against hydrolysis

(Fig 2G). We obtained very similar results when we incubated 40 μM α-synuclein monomers or fibrils with 500 μM polyP, spun down polyP-associated fibrils and measured hydrolyzable polyP in both supernatant and pellet. More than 95% of PPX-hydrolyzable polyP was found in the SN of samples containing soluble α-synuclein monomers. In contrast, about 45% of the total polyP pelleted with α-synuclein fibrils, of which about two-thirds (~130 μM) were resistant towards PPX-mediated hydrolysis (Fig 2H). We concluded from these results that polyP–α-synuclein–fibrils are apparently stable and resistant towards exopolyphosphatase-mediated polyP hydrolysis.

## Extracellular polyP prevents intracellular enrichment of α-synuclein fibrils

Our findings that polyP associates with preformed α-synuclein fibrils and changes their conformation served to explain results of our previous studies, which showed that α-synuclein fibrils lose their cytotoxicity as soon as polyP is added (Cremers et al, 2016). However, they did not explain how polyP is able to protect against amyloid toxicity. We reasoned that polyP might reduce the formation of cytotoxic oligomers by stabilizing the fibrils in a conformation that has previously been shown to be less prone to shedding (Cremers et al, 2016). Alternatively, we considered that binding of polyP to the fibrils might either directly or indirectly

interfere with the membrane association of α-synuclein (van Rooijen et al, 2008; Grey et al, 2011) and/or its cellular uptake (Reyes et al, 2015; Karpowicz et al, 2017). Last, it was also conceivable that polyP binding might increase the turnover of internalized α-synuclein or its sequestration into nontoxic deposits.

To gain insights into the potential mechanism(s) by which polyP protects neuronal cells against amyloid toxicity, we compared uptake and intracellular fate of exogenously added α-synuclein fibrils in the absence and presence of polyP. We labeled α-synuclein with Alexa Fluor 488, formed mature fibrils, pelleted them by centrifugation, and sonicated the fibrils to obtain a mixture of oligomeric species, protofibrils, and short mature fibrils (i.e., α-syn$^{PFF-AF488}$) (Luk et al, 2009; Volpicelli-Daley et al, 2011) (Fig S2A). We confirmed that sonication does not affect the interaction of fibrils with polyP (Fig S2B). We then incubated differentiated SH-SY5Y neuroblastoma cells with α-syn$^{PFF-AF488}$ or freshly prepared fluorescently labeled monomeric α-synuclein (i.e., α-syn$^{mon-AF488}$) at either 4°C or 37°C in the absence or presence of polyP for 24 h and analyzed AF488 fluorescence using confocal microscopy. In the absence of polyP, we detected significant intracellular fluorescence upon incubation of the cells with either α-syn$^{mon-AF488}$ or α-syn$^{PFF-AF488}$ at 37°C but not at 4°C (Fig 3A). Moreover, we noted an apparently

stable association of α-syn$^{PFF-AF488}$ with the cell membrane at both temperatures (Fig 3A), which was confirmed by trypan blue staining (Fig S4A). These results were fully consistent with previous studies, which reported that both monomers and fibrils use a temperature-sensitive endocytic route for their cellular uptake (Rodriguez et al, 2018) and that α-synuclein fibrils stably associate with cell membranes (Karpowicz et al, 2017). Incubation of the cells in the presence of polyP at concentrations ranging from 10 to 500 μM increasingly reduced the intracellular fluorescence signal of α-syn$^{PFF-AF488}$ upon incubation at 37°C as well as the membrane-associated signal upon incubation at either temperature (Figs 3A and S3). This result was distinctly different from monomeric α-syn$^{mon-AF488}$, whose uptake at 37°C was not affected by polyP. These results strongly suggested that polyP negatively influences the membrane association and/or uptake of α-syn$^{PFF-AF488}$.

To test whether intracellular polyP influenced the uptake and/or intracellular foci formation of exogenously added α-syn$^{PFF-AF488}$, we incubated differentiated SH-SY5Y neuroblastoma cells with fluorescently labeled polyP$_{300}$ (i.e., polyP$_{300-AF647}$) for 24 h, washed the cells to remove any exogenous polyP, and analyzed the cells using a confocal microscope. We observed a clear AF647 fluorescence signal in cells incubated with fluorescently labeled

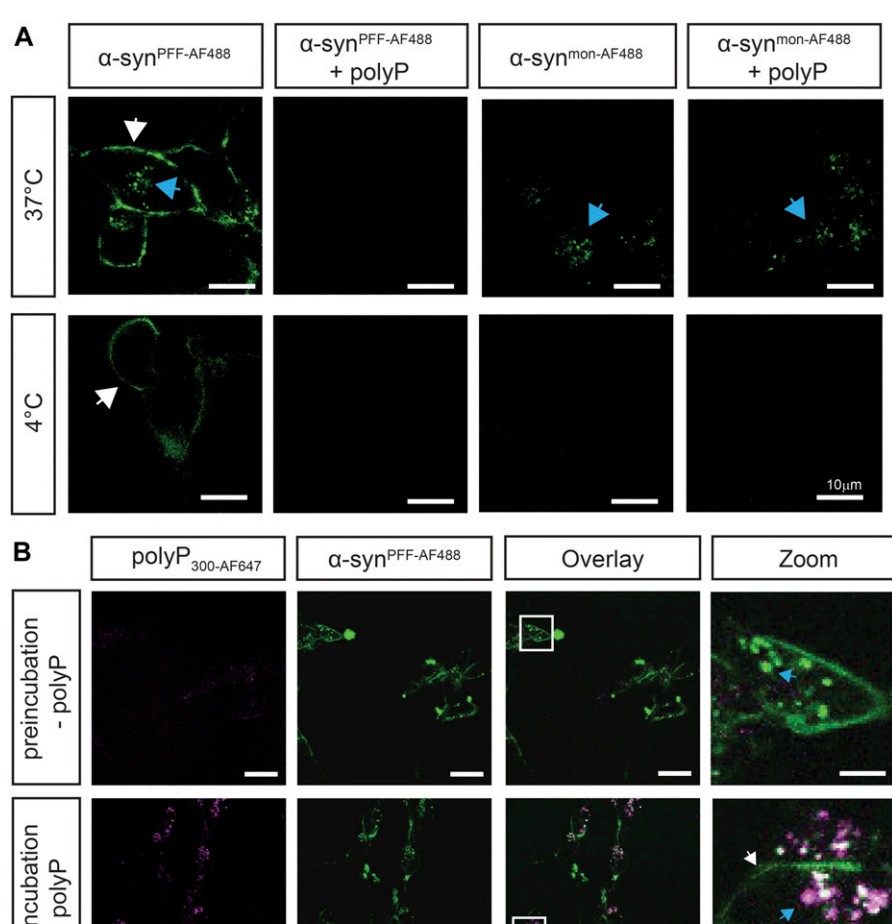

**Figure 3. Extracellular polyP prevents intracellular enrichment of α-synuclein fibrils.**
**(A)** Differentiated SH-SY5Y neuroblastoma cells were incubated with 3 μM freshly purified monomeric α-syn$^{mon-AF488}$ or α-syn$^{PFF-AF488}$ fibrils in the absence or presence of 250 μM polyP$_{300}$ (in P$_i$-units) at either 4°C or 37°C for 3 h. Membrane-associated α-syn$^{AF488}$ is indicated with white arrows, whereas internalized α-syn$^{AF488}$ is indicated with blue arrows. Brightness and contrast adjustments have been equally applied to all images. **(B)** Intracellular enrichment of cells with polyP$_{300}$ neither affects uptake nor turnover of α-syn$^{PFF-AF488}$. Differentiated SH-SY5Y cells were incubated with 250 μM polyP$_{300-AF647}$ at 37°C for 24 h and washed to remove extracellular polyP. Then, 3 μM preformed α-syn$^{PFF-AF488}$ fibrils were added and fluorescence microscopy was conducted after 24 h of incubation. Brightness and contrast adjustments have been equally applied to all images.

polyP but not in our control cells (Fig S4B). This result confirmed previous studies that showed neuronal cells are able to take up and enrich exogenous polyP (Fig S4B) (Holmstrom et al, 2013). When we incubated the polyP-enriched cells with $\alpha$-syn$^{PFF-AF488}$, we observed the same rapid internalization and intracellular enrichment of $\alpha$-syn$^{PFF-AF488}$ fibrils that we found in cells that were not pretreated with polyP (Fig 3B). We concluded from these experiments that polyP needs to be present in the extracellular space to interfere with the uptake of $\alpha$-syn$^{PFF}$, and that intracellular polyP does not substantially affect the fate of internalized $\alpha$-synuclein fibrils. This is despite the fact that we observed a clear co-localization between internalized $\alpha$-syn$^{PFF-AF488}$ and intracellular polyP$_{300-AF647}$ in select intracellular foci, demonstrating that polyP associates with $\alpha$-syn$^{PFF-AF488}$ also in the context of intact cells (Fig 3B, blue arrow).

### PolyP interferes with $\alpha$-syn$^{PFF}$ membrane association

To further investigate the influence of polyP on fibril uptake, we incubated differentiated SH-SY5Y cells with $\alpha$-syn$^{PFF-AF488}$ as before, and either left them untreated or added polyP-defined time points after start of the incubation. We reasoned that determining the effects of polyP on cells that contained both membrane-associated and internalized $\alpha$-syn$^{PFF-AF488}$ would likely reveal at what stage polyP acts. Before the imaging, we washed the cells to remove any unbound $\alpha$-syn$^{PFF}$ and/or polyP. As expected, incubation of SH-SY5Y neuroblastoma cells with $\alpha$-syn$^{PFF-AF488}$ in the absence of polyP revealed a persistent association of labeled $\alpha$-syn$^{PFF-AF488}$ with the cell membrane, and a steady increase in the intracellular fluorescent signal (Fig 4A). When we added polyP to cells that were preincubated with $\alpha$-syn$^{PFF-AF488}$ for 2 h and imaged the samples 30 min later, we observed a significantly reduced signal of membrane-associated $\alpha$-syn$^{PFF-AF488}$ and lower levels of intracellular $\alpha$-syn$^{PFF-AF488}$ compared with the control cells. In the presence of polyP, the fluorescence signals did not significantly change over the next hours of incubation and only a slight increase in the intracellular signal of $\alpha$-syn$^{PFF-AF488}$ was observed after 24 h of incubation. Addition of polyP at later time points (i.e., 4 or 6 h) caused a similar cessation in $\alpha$-syn$^{PFF-AF488}$ uptake and a decrease in cell membrane–associated $\alpha$-syn$^{PFF-AF488}$ signal (Fig 4A). Upon addition of fluorescently labeled polyP$_{300-AF647}$ to cells pretreated with $\alpha$-syn$^{PFF-AF488}$ for 6 h, we found both fluorescence signals to co-localize on the outside of the cells, consistent with the formation of polyP–fibril complexes (Fig 4B). These results strongly suggested that binding of polyP to $\alpha$-synuclein fibrils interferes with the membrane association of $\alpha$-synuclein and, hence, prevents the uptake of fibrils. They also served to explain why the uptake of monomeric $\alpha$-synuclein, which does not stably interact with polyP, is unaffected by the presence of polyP.

Recent studies suggested that one mechanism by which $\alpha$-syn$^{PFF-AF488}$ enter cells is through the interaction with heparin glycan receptors (Ihse et al, 2017), in a mechanism termed micropinocytosis (Nakase et al, 2004). To investigate the possibility that polyP inhibits the uptake of $\alpha$-syn$^{PFF-AF488}$ by generally blocking micropinocytosis, we monitored the influence of polyP on the uptake of the trans-activator of transcription (TAT) protein fused to the fluorescent dye TAMRA (TAT-TAMRA) (AnaSpec). TAT is a small viral protein, which contains the heparan sulfate–binding sequence necessary for its internalization via micropinocytosis (Wadia et al, 2004; Kaplan et al, 2005). We incubated differentiated SH-SY5Y cells with both TAT-5-(and-6)-Carboxytetramethylrhodamine, Succinimidyl Ester (TAMRA) and $\alpha$-syn$^{PFF-AF488}$ either in the absence or in the presence of polyP$_{300}$ and monitored the uptake of both proteins via fluorescence microscopy. In the absence of polyP$_{300}$, we observed signals for both $\alpha$-syn$^{PFF-AF488}$ and TAT-TAMRA in the cells, indicating that both proteins were taken up (Fig 4C). In the presence of polyP, however, we observed only the TAT-TAMRA signal inside the cells (Fig 4C). These results are consistent with the model that polyP selectively prevents the uptake of fibrillary $\alpha$-synuclein without generally interfering with endocytosis mechanisms.

To finally test whether the chain lengths of polyP influences its ability to prevent uptake of $\alpha$-syn$^{PFF-AF488}$, we incubated differentiated SH-SY5Y cells with $\alpha$-syn$^{PFF-AF488}$ as before but added 250 $\mu$M (in P$_i$ units) of either polyP$_{14}$, polyP$_{130}$, or polyP$_{300}$. Analysis of internalized $\alpha$-syn$^{PFF-AF488}$ after 24 h demonstrated that whereas the longer polyP chains completely inhibited the uptake of $\alpha$-syn$^{PFF-AF488}$, presence of polyP$_{14}$ had a much diminished effect on the uptake (Fig 4D). These results were in excellent agreement with our previous competition studies that showed polyP$_{14}$ chains are substantially less effective in binding to $\alpha$-syn$^{PFF-AF488}$ and/or competing with polyP$_{300}$ and excluded that the observed effects are simply due to the presence of densely charged polyanions. Instead, these results provided supportive evidence for the conclusion that the mechanism by which polyP protects neuronal cells against $\alpha$-synuclein toxicity is through its specific interactions with extracellular $\alpha$-synuclein fibrils, effectively preventing their association with the cell membrane and limiting their uptake into neuronal cells.

## Discussion

### Effects of polyP on $\alpha$-synuclein fibril formation and structure

Previous work from our laboratory demonstrated that polyP effectively accelerates fibril formation of disease-related amyloids and protects against amyloid toxicity both in cell culture as well as in disease models of *Caenorhabditis elegans* (Cremers et al, 2016). To gain insights into the mechanism by which polyP exerts these effects, we tested at what stage during the fibril-forming process, polyP acts on $\alpha$-synuclein, one of the major amyloidogenic proteins involved in PD. These studies showed that polyP is unable to accelerate the rate-limiting step of $\alpha$-synuclein fibril formation. Instead, polyP binds to $\alpha$-synuclein species that begin to accumulate at the end of the lag phase and are present throughout the elongation and stationary phase of fibril formation. These results agreed well with previous solution studies, which showed that polyP does not promote the conversion of $\alpha$–helical proteins into $\beta$-sheet structures but instead stabilizes folding intermediates once they have adopted a $\beta$-sheet conformation (Yoo et al, 2018). These results also make physiological sense as they exclude the possibility that simple co-presence of polyP and $\alpha$-synuclein in the same cellular compartment cause fibril formation.

Earlier work on $\alpha$-synuclein has shown that the primary nucleation step involves structural changes within $\alpha$-synuclein monomers and formation of small pre-fibrillar oligomeric intermediates, which are rich in $\beta$-sheet structures yet unable to

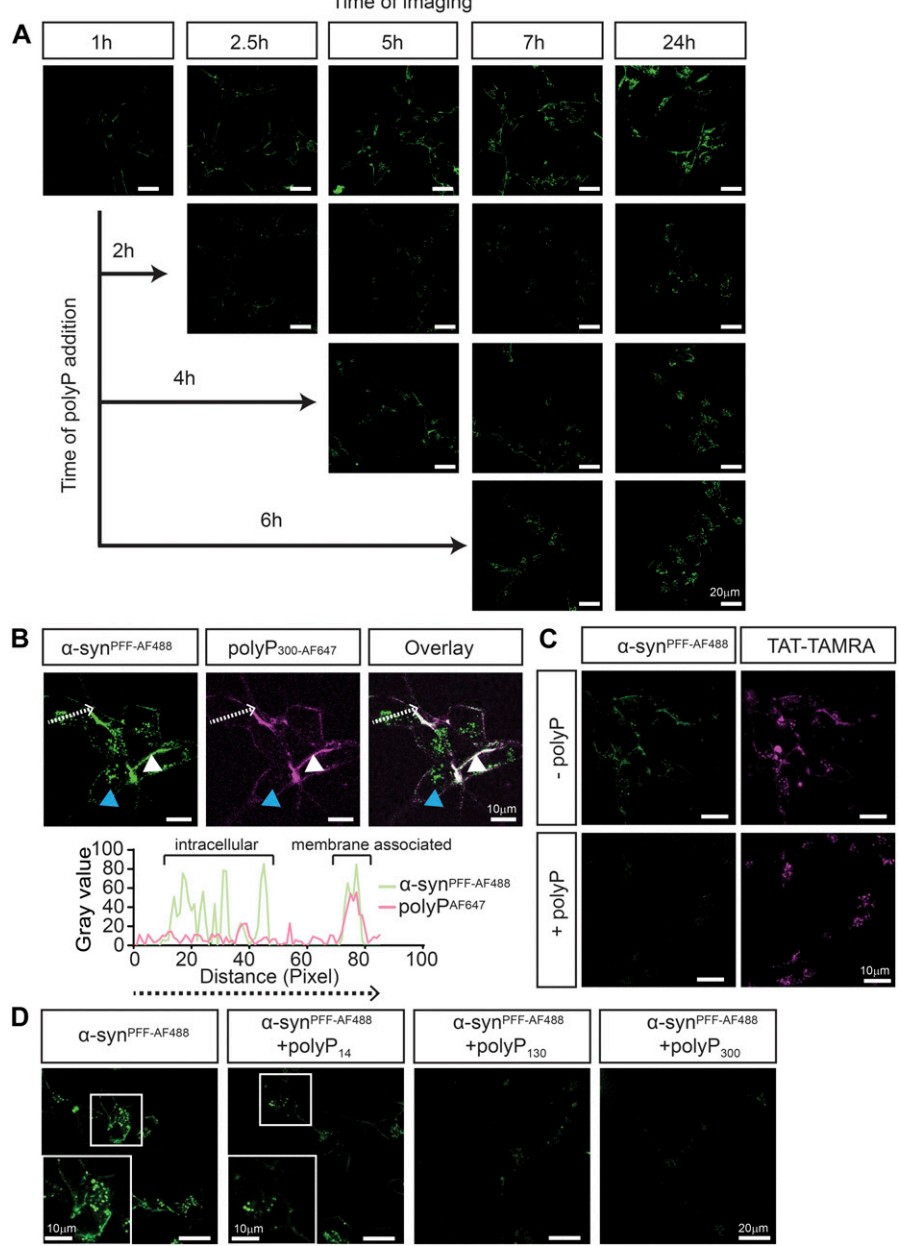

**Figure 4. PolyP prevents the association of α-synuclein fibrils with the membrane.**
**(A)** Uptake of 3 μM preformed α-syn[PFF-AF488] fibrils after 1, 2.5, 5, 7 and 24 h into differentiated SH-SY5Y cells. After 2, 4, or 6 h of incubation, 250 μM polyP[300-AF647] was added, and the uptake of α-syn[PFF-AF488] fibrils was monitored as indicated. Brightness and contrast adjustments have been equally applied to all images. **(B)** Upper panel: differentiated SH-SY5Y cells were incubated with 3 μM preformed α-syn[PFF-AF488] fibrils for 6 h. Then, 250 μM polyP[300-AF647] was added and co-localization of α-syn[PFF-AF488] fibrils and polyP[300-AF647] was determined. The intracellular α-syn[PFF-AF467] signal is indicated with blue arrows, whereas extracellular α-syn[PFF-AF467] is indicated with white arrows. polyP[300-AF647] was only detected on the cell surface. Lower panel: fluorescence signal of α-syn[PFF-AF488] fibrils and polyP[300-AF647] as measured along the white line marked in the upper figure using the plot profile analysis in ImageJ. **(C)** Brightness and contrast have been adjusted to for ideal comparison α-syn[PFF-AF467] and polyP[300-AF647] signal. (C) Differentiated SH-SY5Y cells were incubated with 5 μM TAT-TAMRA and 3 μM α-syn[PFF-AF488] in the presence or absence of 250 μM polyP[300]. After 3 h of incubation, the uptake was monitored. Brightness and contrast adjustments have been equally applied to all images. **(D)** Differentiated SH-SY5Y cells were incubated with α-synuclein fibrils for 24 h in the absence or presence of 250 μM of different chain lengths of polyP. Brightness and contrast adjustments have been equally applied to all images.

increase ThT-fluorescence (Conway et al, 2000a; Krishnan et al, 2003; Mehra et al, 2018). It has been proposed that these oligomers undergo a cooperative conformational change, leading to the formation of ThT-positive protofibrils and fibrils (Mehra et al, 2018). Our findings that polyP binding slightly, yet reproducibly, precedes ThT binding and substantially accelerates the formation of ThT-positive protofibrils and fibrils suggest that polyP serves as a binding scaffold for pre-fibrillar oligomers and increases the cooperativity of fibril formation.

A recently solved cryo-EM structure of mature α-synuclein$_{1–121}$ fibrils revealed that the double-twisted nature of the fibrils results from the association of two protofilaments, which are stabilized by intermolecular salt bridges (Fig S5) (Guerrero-Ferreira et al, 2018).

Moreover, the fibrils are characterized by dense positively charged patches that are located in the vicinity of the interface and run in parallel to the fibril axis. We now hypothesize that binding of the negatively charged polyP chains to such densely positively charged patches that run alongside individual oligomers will support the correct orientation of the oligomers along the fiber axis, hence, nucleate fibril formation. This model would explain why polyP shows very low apparent affinity for soluble α-synuclein monomers and provide some rationale for the very low binding stoichiometry of polyP to α-synuclein, which is a mere 5 P$_i$-units per one α-synuclein monomer. However, future high-resolution structure studies are clearly necessary to answer the important question as to how polyP and α-synuclein fibrils interact.

Our studies showed that polyP does not only change the morphology of α-synuclein fibrils when present during de novo fibril formation but also of mature α-synuclein fibrils. These results agree with recent findings, which suggested that fibrils are intrinsically dynamic and can adopt different conformations over a time scale of weeks to months (Sidhu et al, 2017). Given that the two strands associate via charge–charge interactions (Fig S5), the most obvious explanation is that the negatively charged polyP causes repulsion of the two strands, thereby initiating dissociation. It will be interesting to assess the relative effects of polyP on the fibril morphology of disease-associated mutant, which appears to be more resistant to morphology changes than wild-type α-synuclein. In either case, however, our results suggest that polyP might play a pivotal role as modifier of disease-associated fibrils.

### Mechanistic insights into polyP's protective role against α-synuclein toxicity

The toxicity associated with α-synuclein fibril formation has long been attributed to the cellular accumulation of insoluble fibril deposits (El-Agnaf & Irvine, 2000; Goldberg & Lansbury, 2000). However, increasing evidence now suggests that oligomeric intermediates, which accumulate during amyloid fibril formation, interfere with membrane integrity, mitochondrial activity, and/or other physiologically important functions and elicit the neuro-inflammatory responses associated with the disease (Chen et al, 2015). Similarly, disease progression has also been proposed to be the responsibility of amyloid oligomers, which appear to be able to spread from cell to cell in a prion-like manner (Li et al, 2008; Danzer et al, 2009; Desplats et al, 2009). Rodents that were subjected to α-syn[PFF] injections into the *striatum*, for instance, developed neurodegeneration in the substantia nigra (Luk et al, 2012; Paumier et al, 2015). These results suggested that physiologically relevant amyloid modifiers, such as polyP, which are present in the extracellular space in the brain (Holmstrom et al, 2013) and protect against amyloid-induced cytotoxicity in cell culture models, might be involved in the spreading of the disease. We now revealed that binding of polyP to extracellular α-syn[PFF] decreased the membrane association of α-syn[PFF] and significantly reduced the internalization of α-synuclein fibrils (Fig 5), likely explaining the reason for polyP's cytoprotective effects. We found this effect to be highly specific for amyloid fibrils because neither the uptake of α-synuclein monomers nor of TAT-TAMRA, which, like α-syn[PFF] is internalized via micropinocytosis (Wadia et al, 2004; Kaplan et al, 2005), was affected by the presence of polyP. Moreover, shorter polyP chain lengths, which are much less effective in interacting with α-syn[PFF] compared with longer chains, were found to be also much less effective in preventing the uptake of the fibrils. Finally, we found that intracellular enrichment with polyP had no effect on the amount of internalized α-synuclein fibrils, indicating that polyP blocks the uptake and not the intracellular turnover rate of α-syn[PFF]. These results suggested that the direct interaction between polyP and α-syn[PFF], through alterations in fibril conformations and/or the abundance of negative charges associated with α-syn[PFF], prevents the interactions of α-syn[PFF] fibrils with the negatively charged lipids on the cell membrane and leads to its dissociation from the

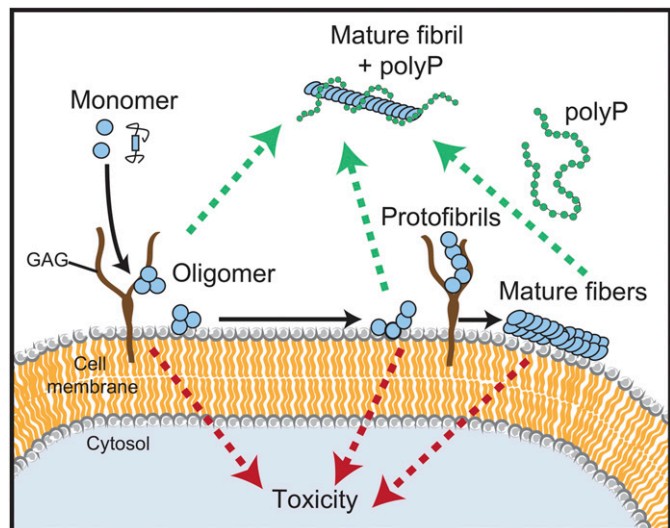

**Figure 5. Model for the influence of polyP on fibril formation, morphology, and uptake.**
PolyP accelerates amyloid fibril formation by nucleating pre-fibrillar oligomers. PolyP-associated fibrils have significantly altered fibril morphology. The interaction of polyP with amyloidogenic α-synuclein interferes with their membrane association and there prevents cellular uptake.

membrane (Ihse et al, 2017). Given that the toxicity of α-synuclein amyloids is attributed to their ability to bind, penetrate, and damage the membrane (van Rooijen et al, 2009; Reynolds et al, 2011), our finding that a physiologically relevant compound affects this process and suggests that polyP might play an important role in the development and/or progression of this disease. Tools need to be developed to quantify, monitor, and manipulate extra- and intracellular polyP levels and test the exciting idea that prevention of the reported age-associated polyP decline in mammalian brains (Lorenz et al, 1997) might serve to delay the onset and/or extent of this devastating disease.

# Materials and Methods

### PolyP preparation

Defined chain length polyP was a kind gift from Dr. Toshikazu Shiba (RegeneTiss). PolyP was labeled with Alexa Fluor 647 as described (Choi et al, 2010). In brief, 125 $\mu$M of polyP$_{300}$ chain was incubated with 2.5 mM Alexa Fluor 647 cadaverine (Life Technologies) and 200 mM 1-ethyl-3- (3-dimethylaminopropyl) carbodiimide (EDAC) (Invitrogen) in water for 1 h at 60°C. The reaction was stopped on ice and labeled polyP$_{300-AF647}$ was separated from free dye and unlabeled polyP via a NAP-5 column (GE Healthcare) that was equilibrated with 40 mM KPi, pH 7.5. The concentration of polyP was determined via a toluidine blue assay (Mullan et al, 2002). In this assay, polyP was mixed with 6 mg/l toluidine blue and the absorbance was measured at 530 and 630 nm. The 530/630 nm absorbance ratio was determined and the concentration was calculated based on a polyP$_{300}$ standard curve.

## Protein purification and labeling of α-synuclein

Alpha-synuclein WT or α-synuclein A90C mutant was purified as described (Jain et al, 2013; Cremers et al, 2016) with slight modifications. In brief, *Escherichia coli* strain BL21 (DE3) containing the α-synuclein–expressing vector pT7-7 was grown in Luria broth, supplemented with 200 $\mu$g/ml ampicillin until $OD_{600}$ of 0.8–1.0 was reached. The protein expression was induced with 0.8 mM IPTG for 4 h. Then, bacteria were harvested at 4,500$g$ for 20 min and 4°C. The pellet was resuspended in 50 ml lysis buffer (10 mM Tris–HCl, pH 8.0, 1 mM EDTA, Roche Complete protease inhibitor cocktail) and the lysate was boiled for 15–20 min. The aggregated proteins were removed by centrifugation at 13,500$g$ for 30 min. Next, 136 $\mu$l/ml of a 10% wt/vol solution streptomycin sulfate solution and 228 $\mu$l/ml glacial acetic acid were added to the supernatant. After another centrifugation step at 13,500$g$ for 30 min, the supernatant was removed and mixed in a 1:1 ratio with saturated ammonium sulfate and incubated stirring at 4°C for 1 h. The mixture was spun down at 13,500$g$ for 30 min and the pellet was resuspended in 10 mM Tris–HCl, pH 7.5. Concentrated NaOH was used to adjust the pH of the suspension to pH 7.5. Afterwards, the protein was dialyzed against 10 mM Tris–HCl, pH 7.5, 50 mM NaCl, filtered, and loaded onto three connected 5 ml HiTrap Q HP columns (GE Healthcare). Washing steps were performed with 10 mM Tris–HCl, pH 7.5, 50 mM NaCl and the protein was eluted with a linear gradient from 50 to 500 mM NaCl. The protein-containing fractions were combined and dialyzed against 50 mM ammonium bicarbonate, pH 7.8. Oligomeric α-synuclein species were removed by filtering the protein through a 50-kD cutoff column (Amicon, Millipore). Aliquots of the protein were prepared, lyophilized, and stored at –80°C. For crosslinking of α-synuclein-A90C with Alexa Fluor 488–maleimide (Invitrogen), 100 $\mu$M of the protein was incubated with 1 mM Tris(2-carboxyethyl) phosphine (Invitrogen) for 30 min at RT protected from light. Alexa Fluor 488–maleimide was added in threefold excess and the mixture was incubated overnight at 4°C. The reaction was stopped by adding 2 mM dithiothreitol. The free dye was removed using an NAP column (GE Healthcare). The concentration of dye and protein were determined by measuring absorbance at 488 and 280 nm, respectively. *Saccharomyces cerevisiae* exopolyphosphate (ScPPX) was purified according to Pokhrel et al (2019) with slight modifications. In brief, MJG317 (BL21/pScPPX2 = *S. cerevisiae* PPX1 in pET-15b) was incubated overnight at 37°C without shaking in LB containing 100 $\mu$g mL$^{-1}$ ampicillin. The next day, the cultures were shaken for about 30 min at 180 rpm at 37°C until they reach an absorbance of 0.4–0.5. Additional 100 $\mu$g mL$^{-1}$ ampicillin and IPTG to a final concentration of 1 mM were added and protein was expressed by incubating the cells for 4 h at 37°C with shaking at 180 rpm. The cells were harvested by centrifuging for 20 min at 4,000 rpm at 4°C. The pellet was resuspended in 50 mM sodium phosphate buffer, 500 mM NaCl, 10 mM imidazole (pH 8) and 1 mg/mL lysozyme, 2 mM MgCl2, and 50 U/mL Benzonase was added. The solution was incubated for 30 min on ice to digest nucleotides. Cell lyses was performed via sonication with two cycles of 50% power pulsing 5 s on and 5 s off for 2 min with 2 min rest between cycles. The protein lysate was centrifuged to remove cell debris for 20 min at 20,000$g$ at 4°C and loaded onto a nickel-charged chelating column. After washing with 50 mM sodium phosphate buffer, 0.5 M

NaCl and 10 mM imidazole (pH 8), and 50 mM sodium phosphate buffer, 0.5 M NaCl and 20 mM imidazole (pH 8), the samples were eluted with 50 mM sodium phosphate buffer, 0.5 M NaCl and 0.5 M imidazole (pH 8). Fractions containing ScPPX were pooled and dialyzed twice against 2 liters of 20 mM Tris–HCl (pH 7.5), 50 mM KCl, 30% (vol/vol) glycerol. Precipitated protein was removed via centrifugation for 20 min at 20,000$g$ at 4°C, 50% glycerol was added, and the protein was stored at –80°C.

## Preparation of fluorescently labeled α-syn$^{PFF}$

To generate α-syn$^{PFF-AF488}$, 760 $\mu$M freshly purified α-synuclein monomers were incubated with 40 $\mu$M labeled α-synuclein-AF488 in 40 mM KPi and 50 mM KCl, pH 7.5, for 24 h at 37°C under continuous shaking using two 2-mm borosilicate glass beads (Sigma-Aldrich) in clear 96-well polystyrene microplates (Corning) (Giehm & Otzen, 2010). Samples from the 96-well plate were combined in Eppendorf tubes and collected via centrifugation at 20,000$g$ for 20 min at RT, and the pellets were washed twice with 40 mM KPi and 50 mM KCl, pH 7.5, to remove smaller oligomers. After the final spin, the pellets were resuspended in 40 mM KPi and 50 mM KCl, pH 7.5, and sonicated 3 × 5 s on ice with an amplitude of 50%. The concentration of fibrils was determined by incubating a small aliquot of α-syn$^{PFF-AF488}$ in 8 M urea and 20 mM Tris, pH 7.5, measuring the absorbance at 280 nm and calculating the concentration with the extinction coefficient of 5,960 liters mol$^{-1}$ cm$^{-1}$. Aliquots were taken and stored at –80°C.

## Thioflavin T fluorescence and FP measurements

Freshly purified α-synuclein monomers (concentrations provided in the respective figure legends) were incubated with 10 $\mu$M thioflavin T (ThT; Sigma-Aldrich) in 40 mM KPi and 50 mM KCl, pH 7.5, at 37°C and two 2-mm borosilicate glass beads (Sigma-Aldrich) in the absence or presence of polyP$_{14}$ or polyP$_{300}$ (given in $P_i$ units). For ThT measurements, the samples were pipetted into black 96-well polystyrene microplates with clear bottoms (Greiners). ThT fluorescence was detected in 10-min intervals using a Synergy HTX MultiMode Microplate Reader (Biotec) using an excitation of 440 nm, emission of 485 nm, and a gain of 35. To monitor the binding of polyP to α-synuclein during fibril formation, the samples were pipetted into 96-well polystyrene microplates with clear bottoms (Greiners). FP was measured in a Tecan Infinite M1000 microplate reader, using an excitation of 635 nm and an emission of 675 nm. Measurements were taken in 10-min intervals.

## PolyP binding and competition assays using anisotropy measurements

Anisotropy measurements were conducted in the Varian Cary Eclipse Fluorescence Spectrophotometer, using an excitation of 640 nm and an emission of 675 nm (photomultiplier tube value set between 50 and 100). Samples containing 50 $\mu$M polyP-AF647 in 40 mM KPi and 50 mM KCl, pH 7.5, at 37°C. At the indicated time points, 30 $\mu$M of α-synuclein monomers or α-synuclein fibrils were added and anisotropy was further monitored over time. For competition experiments, α-synuclein fibrils were formed in the presence of polyP$_{300-AF647}$ as before. At

defined time points, unlabeled polyP$_{14}$ or polyP$_{300}$ was added, and the anisotropy signal was monitored over time.

## Negative stain of fibrils and TEM analysis

To form fibrils for TEM analysis, 300 $\mu$M freshly prepared $\alpha$-synuclein monomers were incubated either in the absence of polyP (i.e., $\alpha$-syn$^{alone}$) or in the presence of 7.5 mM (per Pi) polyP$_{300}$ (i.e., $\alpha$-syn$^{polyP}$) for 24 h at 37°C with 2-mm borosilicate glass beads under continuous shaking. Alpha-syn$^{alone}$ fibrils were then either left untreated or were incubated with 7.5 mM polyP$_{300}$ for 20 min (i.e., $\alpha$-syn$^{alone \rightarrow polyP}$). The samples were negatively stained with 0.75% uranyl formate (pH 5.5–6.0) on thin amorphous carbon-layered 400-mesh copper grids (Pelco) in a procedure according to Ohi et al (2004). Briefly, 5 $\mu$l of the sample was applied onto the grid and left for 3 min before removing it with Whatman paper. The grid was washed twice with 5 $\mu$l ddH$_2$O, followed by three applications of 5 $\mu$l uranyl formate. The liquid was removed using a vacuum. The grids were imaged at RT using a Fei Tecnai 12 microscope operating at 120 kV. Images were acquired on a US 4,000 CCD camera at 66873× resulting in a sampling of 2.21 Å/pixel. About 30–40 individual $\alpha$-synuclein filaments were selected across 10 micrographs of each sample and the filament widths were determined using the micrograph dimensions as a reference. Pixel widths were converted into angstroms using the program ImageJ.

## X-ray fiber diffraction

$\alpha$-synuclein fibrils were grown with and without polyP as described above. Before orientation for diffraction, 1–2 ml of a solution containing 100 $\mu$M $\alpha$-synuclein fibrils were washed three times with 10 mM Tris, pH 7. The fibrils were then pelleted by centrifugation (15,000$g$, 5 min). The supernatant was removed and the pellet was resuspended in 5–10 $\mu$l 10 mM Tris, pH 7.0. 5 $\mu$l of the fibril pellet was then placed between two fire-polished silanized glass capillaries and oriented by air-drying. The glass capillaries with the aligned fibrils were mounted on a brass pin. Diffraction patterns were recorded using 1.13 Å X-rays produced by a 21-ID-D beamline, Argonne Photon Source. All patterns were collected at a distance of 200 mm and analyzed using the Adxv software package (Arvai, 2015).

## PolyP concentration determination using the molybdate assay

40 $\mu$M of $\alpha$-synuclein monomers or fibrils, prepared in 40 mM Hepes, pH 7.5, and 50 mM KCl, were incubated with the indicated concentrations of polyP$_{300}$ for 10 min at RT in a clear 96-well plate (Corning). The samples were either used directly or spun down at 20,000$g$ for 20 min at RT to remove any unbound polyP. The pellets were resuspended in 40 mM Hepes (pH 7.5) and 50 mM KCl. Next, 8 $\mu$g/ml *Saccharomyces cerevisiae* exopolyphosphate (ScPPX) and 1 mM MgCl$_2$ was added to each sample and the incubation was continued for 105 min (for spin down) or 120 min (for titration) at RT. To stop the reaction and detect P$_i$, 25 $\mu$l of a detection solution containing 600 mM H$_2$SO$_4$, 88 mM ascorbic acid, 0.6 mM potassium antimony tartrate, and 2.4 mM ammonium heptamolybdate was added (Christ & Blank, 2018; Pokhrel et al, 2019). The reactions were developed for 30 min. Then, the precipitated proteins were resolubilized

with 100 $\mu$l of 1 M NaOH, and the absorbance was measured at 882 nm using a Tecan M1000 plate reader. The free phosphate concentration was determined with a standard curve of sodium phosphate, which was prepared in parallel with each experiment. After the spun down, the phosphate measured in the supernatant was considered free, and the phosphate measured in the pellet was considered bound and unprotected. The bound and protected fraction of phosphate was calculated as total polyphosphate (measured in parallel) minus supernatant phosphate minus pellet phosphate.

## Cell culture experiments and microscopy

Human neuroblastoma cells, SH-SY5Y cells (ATCC CRL-2266), were cultured in Dulbecco's Modified Eagle Medium/Nutrient Mixture F-12 (Thermo Fisher Scientific) medium supplemented with 10% (vol/vol) heat-inactivated fetal bovine serum (Sigma-Aldrich), 1% (wt/vol) penicillin/streptomycin (Life Technologies) at 37°C, and 5% CO$_2$. The media was changed every 2–3 d and the cells were split 1–2 times per week. For microscopy experiments, 60,000 cells/ml were seeded in eight-well Nunc Lab-Tek II Chambered Coverglass (Thermo Fisher Scientific) and differentiated for 5–7 d by adding 10 $\mu$M all-trans retinoic acid (Sigma-Aldrich) every other day. The differentiated cells were incubated with 3 $\mu$M $\alpha$-syn$^{PFF-AF488}$ in the presence or absence of the indicated concentrations of polyP at either 37°C or 4°C for the indicated times. Before the imaging, the media was exchanged to DMEM/F12 without phenol red (Thermo Fisher Scientific), supplemented with 10% (vol/vol) heat-inactivated fetal bovine serum (Sigma-Aldrich), and 1% (wt/vol) penicillin/streptomycin (Life Technologies). The cells were imaged using a Leica SP8 high-resolution microscope. To distinguish between the inside and outside signals, the cells were treated the same way, but 0.05% of the membrane-impermeable dye trypan blue was used for 15 s before the imaging to quench extracellular fluorescence (Karpowicz et al, 2017). To enrich for endogenous polyP, SH-SY5Y cells were seeded and differentiated as described above. Once differentiated, the cells were either left untreated or incubated with 250 $\mu$M polyP-AF647 (per Pi) for 24 h. Subsequently, fresh medium was added to the cells for 6 h. Afterwards, the cells were incubated with 3 $\mu$M $\alpha$-syn$^{PFF-AF488}$ for 24 h. As before, the media was changed before imaging and the cells were imaged using Leica SP8 high-resolution microscope. To test the influence of polyP during the $\alpha$-syn$^{PFF-AF488}$ uptake, differentiated SH-SY5Y cells were incubated with 3 $\mu$M $\alpha$-syn$^{PFF-AF488}$ at 37°C. After 2, 4, or 6 h, 250 $\mu$M polyP-AF647 was added to the cells. The cells were imaged at time points 1, 2.5, 5, 7, and 24 h. To test for co-localization of $\alpha$-syn$^{PFF-AF488}$ and polyP-AF647, the cells were incubated with 3 $\mu$M $\alpha$-syn$^{PFF-AF488}$ at 37°C. After 6 h, 250 $\mu$M polyP-AF647 was added and the cells were imaged after 7 h. To monitor the influence of polyP on the uptake of TAT, differentiated SH-SY5Y cells were incubated with 5 $\mu$M TAT-TAMRA (AnaSpec) and 3 $\mu$M $\alpha$-syn$^{PFF-AF488}$ either in the presence or in the absence of 250 $\mu$M polyP. After 3 h of incubation, the cells were imaged with a Leica SP8 high-resolution microscope.

## Statistics

Two-tailed $t$ tests were performed when two groups were compared. One-way ANOVA was performed when comparing more than

two groups. *P*-values under 0.05 were considered significant. All data in bar charts are displayed as mean ± SD. Replicate numbers (n) are listed in each figure legend. Prism 7.04 (GraphPad) was used to perform statistical analysis. ANOVA analysis in Fig 2B shows an F value of 8.435 and a degree of freedom of 3.

## Data Availability

The datasets generated are available from the corresponding author upon request.

## Supplementary Information

## Acknowledgements

Defined-length polyP chains were kindly provided by T. Shiba (Regenetiss) and the Morrissey lab. We thank the American Type Culture Collection for providing us with SH-SY5Y cells. This work was funded by the NIH grants GM122506 to U Jakob and a grant from the American Parkinson Disease Association to MI Ivanova and A Sutter. J Lempart was funded by a scholarship from the Boehringer Ingelheim Foundation.

### Author Contribution

J Lempart: conceptualization, data curation, formal analysis, supervision, funding acquisition, project administration, and writing—original draft, review, and editing.
E Tse: data curation, formal analysis, investigation, and methodology.
JA Lauer: investigation and methodology.
MI Ivanova: data curation, formal analysis, and supervision.
A Sutter: data curation, formal analysis, investigation, and methodology.
N Yoo: data curation, formal analysis, investigation, and methodology.
P Huettemann: investigation and methodology.
D Southworth: conceptualization, data curation, formal analysis, and project administration.
U Jakob: conceptualization, supervision, funding acquisition, investigation, project administration, and writing—review and editing.

### Conflict of Interest Statement

The authors declare that they have no conflict of interest.

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
