## [Reviewer comments · Life Science Alliance]

Life Science Alliance

Mechanistic Insights into the Protective Roles of Polyphosphate Against Amyloid Cytotoxicity

Justine Lempart, Eric Tse, James Lauer, Magdalena Ivanova, Alexandra Sutter, Nicholas Yoo, Philipp Huettemann, Daniel Southworth, and Ursula Jakob

DOI: <https://doi.org/10.26508/lsa.201900486>

Corresponding author(s): Ursula Jakob, University of Michigan

Review Timeline:

Submission Date:	2019-07-16
Editorial Decision:	2019-08-15
Revision Received:	2019-09-04
Editorial Decision:	2019-09-09
Revision Received:	2019-09-09
Accepted:	2019-09-10

Scientific Editor: Andrea Leibfried

Transaction Report:

August 15, 2019

Re: Life Science Alliance manuscript #LSA-2019-00486

Prof. Ursula Jakob
University of Michigan
Department of Molecular, Cellular and Developmental Biology
1105 N-University Ave.
Ann Arbor, MI 48109

Dear Dr. Jakob,

Thank you for submitting your manuscript entitled "Mechanistic Insights into the Protective Roles of Polyphosphate Against Amyloid Cytotoxicity" to Life Science Alliance. The manuscript was assessed by expert reviewers, whose comments are appended to this letter.

As you will see, the reviewers appreciate your analyses and the high quality of the data provided. They give constructive input on how to further strengthen your work, and we would thus like to invite you to submit a revised version, addressing the points put forward by reviewer #2 and #3. This seems rather straightforward, but please do get in touch with us in case you would like to discuss individual revision points further.

Thank you for this interesting contribution to Life Science Alliance. We are looking forward to receiving your revised manuscript.

Sincerely,

Andrea Leibfried, PhD
Executive Editor

Life Science Alliance
Meyerhofstr. 1
69117 Heidelberg, Germany
t +49 6221 8891 502
e a.leibfried@life-science-alliance.org
www.life-science-alliance.org

B. MANUSCRIPT ORGANIZATION AND FORMATTING:

Reviewer #1 (Comments to the Authors (Required)):

This is an exceptionally interesting and well written manuscript defining the protective role of polyP in α -synuclein toxicity, a protein linked to Parkinson's Disease. By combining cell biology experiments in neurons with biophysical and biochemical analyses, the authors provide mechanistic insights into the physical and cellular basis of protection and define a compelling mechanism of action. The authors show that polyP does not interact with α -synuclein monomers, but rather interacts with oligomers and accelerates fibril formation. One fascinating observation is that polyP

alters fibril morphology, and effectively reduces the ability of α -synuclein fibrils to interact with cell membranes, which is essential for both toxicity and for the spread of the disease state throughout the brain. The clearly explained and well controlled experiments support a well justified model for how polyP affects α -synuclein structure, aggregation and toxicity in neurons. The fact that polyP is a physiologically relevant protective agent enhances the overall interest of the study.

Reviewer #2 (Comments to the Authors (Required)):

Lempar et al find that poly-phosphate (polyp) changes the morphology of fibrils of alpha-synuclein (aSyn) and prevent cell uptake by micropinocytosis. The study reveals unexpected findings that conceptually advances our understanding of the modulation of neurotoxic fibrils structures. It is spectacular that polyphosphate even disassembles pre-formed aSyn into single protofilaments. The findings are of interest for a broad readership. Overall the data back up the conclusions.

Before publication, the authors should implement the following suggestions.

1. The authors need to provide a titration of polyP, both for the in vitro data (Fig 1) and for the microscopic data (Figs 3 and 4). They use either 500 μ M or 250 μ M. They mention in the discussion a stoichiometry of 5 phosphate units per aSyn. To make the point they should add for each of the basic experiments one titration, which allows them to continue for all other panels with the optimal concentration.
2. In Fig. 1, the authors conclude from the polarisation data that polyP binding to aSyn precedes fibril growth, as judged by ThT incorporation. Normalisation in the right panel is based on the assumption that saturation of the ThT signal and the polarisation signal meet saturation at the same point. The authors did not show that this is the case. In fact, polarisation signals meet a maximum value if the particle reaches a certain size. This range is bigger for a flexible entity such as polyp compared to a globular protein. However, it still may be the case that the signal saturates earlier than the ThT signal. If the authors want to draw conclusions out of the difference (as they currently do) they would need to prove this point. Otherwise, it may be more telling to monitor the rate of signal increase instead of normalising the data to the saturation signal.
3. The authors show only one fibril for each of the conditions in Fig. 2b. Please provide a more quantitative assessment. Do these images reflect 100% of (how many) fibrils analysed?
4. The authors discuss the mechanism in view of the cryo-EM structure. A visualisation of the argument using the pdb file of the structure will improve the ability of readers to follow the arguments of the authors.
5. The authors do not provide a mechanistic explanation how polyp disassembled the aSyn filaments. To me, a likely explanation is that binding of a negatively charged polymer causes repulsion of the two strands, thereby initiating dissociation. The same effect may also play a role for preventing binding of aSyn to membranes, as observed by the authors. At present the authors do not clearly offer a molecular explanation. They may want to discuss this proposal or offer alternative explanations.
6. Some paragraphs are very long, running over more than a full page. Breaking such them up will improve readability.

Reviewer #3 (Comments to the Authors (Required)):

This manuscript continues the investigation by the Jakob group into the interaction of amyloidogenic proteins (this time focussing on alpha-synuclein (AS)) with polyphosphate.

Overall the manuscript is extremely well written with a clear narrative that explains the logic behind each experiment and how the results were interpreted. The data begin to reveal how polyphosphate increases amyloidogenesis and yet decrease cellular uptake.

The rationale for concluding that "polyp does not affect rate-limiting step of fibril formation" (p6) is very unclear. The authors show that the lag time is reduced from 6 to 1.5 hr in the presence of polyp and that adding the polyp induces aggregation with a shorter lag time as the incubation time of AS prior to polyp addition increases. Figure 1d shows that in the absence of PolyP, the lag time decreases as [AS] increases. These observations suggest that PolyP mimics the effects of increasing [AS] which presumably increases the [oligo]. Are the authors saying that the decrease in time taken for a response to be seen after addition of PolyP shows that oligo formation has to occur and this is not affected by PolyP? Doesn't the observation that lag time is reduced in the presence of polyP has to lead to the conclusion that PolyP does affect the rate limiting step? possibly by shifting mono - oligo equilibrium to the right?

How does polyP alter the conformation of pre-formed fibres, given that it appears to not be tightly bound?

The conclusion of the work on understanding the resistance of AS fibre-bound polyp to phosphatase needs further clarification. What does "We concluded from these results that AS fibrils resist conformational rearrangements by interfering with exopolyphatase-mediated polyp hydrolysis" mean? How does this give any insight into the mechanism of binding, acceleration of aggregation or reduced cell membrane binding?

In the discussion, the authors suggest that polyp interacts with positively charged patch in the vicinity of the protofibril interface. There have been several AS fibril structures published recently with quite distinct interfaces (e.g Li et al Nat Commun 2018, Guerrero-Ferreira eLife 2018). Do the conditions used in this study mimic those where the positive patch forms the interface?

Reviewer #1 (Comments to the Authors (Required)):

This is an exceptionally interesting and well written manuscript defining the protective role of polyP in α -synuclein toxicity, a protein linked to Parkinson's Disease. By combining cell biology experiments in neurons with biophysical and biochemical analyses, the authors provide mechanistic insights into the physical and cellular basis of protection and define a compelling mechanism of action. The authors show that polyP does not interact with α -synuclein monomers, but rather interacts with oligomers and accelerates fibril formation. One fascinating observation is that polyP alters fibril morphology, and effectively reduces the ability of α -synuclein fibrils to interact with cell membranes, which is essential for both toxicity and for the spread of the disease state throughout the brain. The clearly explained and well controlled experiments support a well justified model for how polyP affects α -synuclein structure, aggregation and toxicity in neurons. The fact that polyP is a physiologically relevant protective agent enhances the overall interest of the study.

We very much appreciate these very supportive comments (a rare treat).

Reviewer #2 (Comments to the Authors (Required)):

Lempart et al find that poly-phosphate (polyp) changes the morphology of fibrils of alpha-synuclein (aSyn) and prevent cell uptake by micropinocytosis. The study reveals unexpected findings that conceptually advances our understanding of the modulation of neurotoxic fibrils structures. It is spectacular that polyphosphate even disassembles pre-formed aSyn into single protofilaments. The findings are of interest for a broad readership. Overall the data back up the conclusions.

We thank the reviewer for the overall positive assessment of our ms and the very helpful suggestions.

Before publication, the authors should implement the following suggestions.

1. The authors need to provide a titration of polyP, both for the *in vitro* data (Fig 1) and for the microscopic data (Figs 3 and 4). They use either 500 μ M or 250 μ M. They mention in the discussion a stoichiometry of 5 phosphate units per aSyn. To make the point they should add for each of the basic experiments one titration, which allows them to continue for all other panels with the optimal concentration.

We thank the reviewer for this suggestion. We have now referred to our previously published titration experiments (Cremers et al, Mol. Cell 2016) as reference for the *in vitro* data and have conducted additional titration experiments as reference for the *in vivo* data. The new figure can be found in Supplement Fig. 3.

Page 4: "Consistent with previous polyP titrations (Cremers et al, 2016), we found that the presence of 500 μ M polyP substantially accelerates the fibril-forming process of α -synuclein, both by shortening the lag phase and by increasing the rate of fibril growth (Fig. 1b, c)."

Page 8: "Incubation of the cells in the presence of polyP at concentrations ranging from 10-500 μ M increasingly reduced the intracellular fluorescence signal of α -syn^{PFF-AF488} upon incubation at 37°C as well as the membrane-associated signal upon incubation at either temperature (Fig. 3a, Supplemental Fig. 3)."

2. In Fig. 1, the authors conclude from the polarisation data that polyP binding to aSyn precedes fibril growth, as judged by ThT incorporation. Normalisation in the right panel is based on the assumption that saturation of the ThT signal and the polarisation signal meet saturation at the same point. The authors did not show that this is the case. In fact, polarisation signals meet a maximum value if the particle reaches a certain size. This range is bigger for a flexible entity such as polyp compared to a globular protein. However, it still may be the case that the signal saturates earlier than the ThT signal. If the authors want to draw conclusions out of the difference (as they currently do) they would need to prove this point. Otherwise, it may be more telling to monitor the rate of signal increase instead of normalising the data to the saturation signal.

We thank the reviewer and agree that we cannot conclude that the maximum signal in the polarization assay is indeed the maximal signal or just the upper limit of our detection. We have now, as suggested, focused our analysis on the initial lag phase and the growth phase, and adjusted our data

analysis and discussion accordingly.

Page 4: "Since we could not exclude that the plateau was not due to the size of the polyP-fibril complexes reaching the upper limit of our polarization measurements, we only directly compared the first 10 h of ThT binding and anisotropy measurements. Overlay of the normalized data revealed that while the lag phase in the FP-measurement was slightly shorter than the lag phase in the ThT measurements, the rate of signal increase was very similar (Fig. 1b, right panel)."

3. The authors show only one fibril for each of the conditions in Fig. 2b. Please provide a more quantitative assessment. Do these images reflect 100% of (how many) fibrils analysed?

We apologize for not making this point clearer. We have analyzed 41 (polyP^{alone}), 38 (polyP^{alone} → polyP) and 32 (polyP^{polyP}) fibers of each sample and have added representative fibers in our new Supplemental Figure 1.

Page 25, figure legend 2: "Alpha-synuclein fibrils formed in the presence of 7.5 mM polyP₃₀₀ were used as control (α-syn^{polyP}). Quantitative analysis of fibril width was based on 10 individual micrographs and 32-41 α-syn filaments per condition (additional images of representative fibers can be found in Supplemental Fig. 1). Statistical analysis was prepared with ONE-way Anova (****; p-value <0.0001)."

4. The authors discuss the mechanism in view of the cryo-EM structure. A visualisation of the argument using the pdb file of the structure will improve the ability of readers to follow the arguments of the authors.

We fully agree and have now added the structure of α-synuclein fibrils and the localization of charged side chains in supplemental Fig. 5.

Page 11: "A recently solved cryo-EM structure of mature α-synuclein₁₋₁₂₁ fibrils revealed that the double-twisted nature of the fibrils results from the association of two protofilaments, which are stabilized by intermolecular salt bridges (Supplemental Fig. 5) (Guerrero-Ferreira et al, 2018)."

5. The authors do not provide a mechanistic explanation how polyP disassembled the aSyn filaments. To me, a likely explanation is that binding of a negatively charged polymer causes repulsion of the two strands, thereby initiating dissociation. The same effect may also play a role for preventing binding of aSyn to membranes, as observed by the authors. At present the authors do not clearly offer a molecular explanation. They may want to discuss this proposal or offer alternative explanations.

We fully concur with this conclusion and have added additional discussion.

Page: 10 "Given that the two strands associate via charge-charge interactions (Supplemental Fig. 5), the most obvious explanation is that the negatively charged polyP causes repulsion of the two strands, thereby initiating dissociation."

6. Some paragraphs are very long, running over more than a full page. Breaking such them up will improve readability.

We have reworked our ms accordingly.

Reviewer #3 (Comments to the Authors (Required)):

This manuscript continues the investigation by the Jakob group into the interaction of amyloidogenic proteins (this time focussing on alpha-synuclein (AS)) with polyphosphate. Overall the manuscript is extremely well written with a clear narrative that explains the logic behind each experiment and how the results were interpreted. The data begin to reveal how polyphosphate increases amyloidogenesis and yet decrease cellular uptake.

We appreciate the positive evaluation and helpful comments.

The rationale for concluding that "polyP does not affect rate-limiting step of fibril formation" (p6) is very unclear. The authors show that the lag time is reduced from 6 to 1.5 hr in the presence of polyP and that adding the polyP induces aggregation with a shorter lag time as the incubation time of AS prior to polyP addition increases. Figure 1d shows that in the absence of PolyP, the lag time decreases as [AS] increases. These observations suggest that PolyP mimics the effects of increasing [AS] which presumably increases the [oligo]. Are the authors saying that the decrease in time taken for a response to be seen after addition of PolyP shows that oligo formation has to occur and this is not affected by PolyP? Doesn't the observation that lag time is reduced in the presence of polyP has to lead to the conclusion that PolyP does affect the rate limiting step? possibly by shifting mono - oligo equilibrium to the right?

We absolutely agree with this conclusion but must assume an even earlier step, which cannot be overcome by increasing the concentration of the α -synuclein and which cannot be further accelerated by the addition of polyP. We assume that this step is on the level of the monomers, but have no experimental evidence at this point.

How does polyP alter the conformation of pre-formed fibres, given that it appears to not be tightly bound?

This is an excellent question and one that we are currently trying to figure out. Based on our results, we must assume that the fibers are (at least) locally metastable and allow association with polyP, which, in turn, causes conformational rearrangements and shifts the equilibrium to the altered state. We are very interested to test whether degradation of polyP reverses the conformational changes, but were unable to test this idea because of our inability to degrade polyP that is in complex with α -synuclein fibrils.

The conclusion of the work on understanding the resistance of AS fibre-bound polyP to phosphatase needs further clarification. What does "We concluded from these results that AS fibrils resist conformational rearrangements by interfering with exopolyphatase-mediated polyP hydrolysis" mean? How does this give any insight into the mechanism of binding, acceleration of aggregation or reduced cell membrane binding?

We agree that this conclusion is overreaching and have adjusted the statement accordingly.

Page 7: We concluded from these results that polyP- α -synuclein-fibrils are apparently stable, and resistant towards exopolyphosphatase-mediated polyP hydrolysis.

In the discussion, the authors suggest that polyP interacts with positively charged patch in the vicinity of the protofibril interface. There have been several AS fibril structures published recently with quite distinct interfaces (e.g Li et al Nat Commun 2018, Guerrero-Ferreira eLife 2018). Do the conditions used in this study mimic those where the positive patch forms the interface?

The experimental conditions used in the work of Guerrero-Ferreira that lead to structural analysis of the α -synuclein fibril are comparable to our fiber formation conditions. Guerrero-Ferreira utilizes similar concentrations, shaking and temperature conditions. Our TEM images of α -synuclein fibers are in accordance with their findings. We have discussed our identified morphology in context of the published PDB structure which is now included in Supplemental Fig. 5.

September 9, 2019

RE: Life Science Alliance Manuscript #LSA-2019-00486R

Prof. Ursula Jakob
University of Michigan
Department of Molecular, Cellular and Developmental Biology
1105 N-University Ave.
Ann Arbor, MI 48109

Dear Dr. Jakob,

Thank you for submitting your revised manuscript entitled "Mechanistic Insights into the Protective Roles of Polyphosphate Against Amyloid Cytotoxicity". I appreciate the introduced changes and would thus be happy to publish your paper in Life Science Alliance. Please link your ORCID iD to your profile in our submission system and fill in the electronic license to publish form. You will see that your manuscript number changes to LSA-2019-00486RR, please make sure to move all files to that new submission number.

A. FINAL FILES:

B. MANUSCRIPT ORGANIZATION AND FORMATTING:

Sincerely,

September 10, 2019

RE: Life Science Alliance Manuscript #LSA-2019-00486RR

Prof. Ursula Jakob
University of Michigan
Department of Molecular, Cellular and Developmental Biology
1105 N-University Ave.
Ann Arbor, MI 48109

Dear Dr. Jakob,

Thank you for submitting your Research Article entitled "Mechanistic Insights into the Protective Roles of Polyphosphate Against Amyloid Cytotoxicity". It is a pleasure to let you know that your manuscript is now accepted for publication in Life Science Alliance. Congratulations on this interesting work.

DISTRIBUTION OF MATERIALS:

Again, congratulations on a very nice paper. I hope you found the review process to be constructive and are pleased with how the manuscript was handled editorially. We look forward to future exciting submissions from your lab.

Sincerely,
